# Chemotherapeutic Drugs Inhibiting Topoisomerase 1 Activity Impede Cytokine-Induced and NF-κB p65-Regulated Gene Expression

**DOI:** 10.3390/cancers11060883

**Published:** 2019-06-25

**Authors:** Tabea Riedlinger, Marek Bartkuhn, Tobias Zimmermann, Sandra B. Hake, Andrea Nist, Thorsten Stiewe, Michael Kracht, M. Lienhard Schmitz

**Affiliations:** 1Institute of Biochemistry, Justus Liebig University, D-35392 Giessen, Germany; Tabea.Riedlinger@biochemie.med.uni-giessen.de; 2Institute for Genetics, Justus-Liebig University Giessen, 35392 Giessen, Germany; marek.bartkuhn@gen.bio.uni-giessen.de (M.B.); Sandra.Hake@gen.bio.uni-giessen.de (S.B.H.); 3Bioinformatics and Systems Biology, University of Giessen, Heinrich-Buff-Ring 58–62, 35392 Giessen, Germany; tobias.zimmermann@computational.bio.uni-giessen.de; 4Genomics Core Facility and Institute of Molecular Oncology, Philipps University Marburg, D-35043 Marburg, Germany; andrea.nist@imt.uni-marburg.de (A.N.); stiewe@uni-marburg.de (T.S.); 5Rudolf-Buchheim-Institute of Pharmacology, Justus Liebig University, D-35392 Giessen, Germany; Michael.Kracht@pharma.med.uni-giessen.de

**Keywords:** cytokines, chemotherapy, TOP1 inhibitors, NF-κB, side-effects, infection

## Abstract

Inhibitors of DNA topoisomerase I (TOP1), an enzyme relieving torsional stress of DNA by generating transient single-strand breaks, are clinically used to treat ovarian, small cell lung and cervical cancer. As torsional stress is generated during transcription by progression of RNA polymerase II through the transcribed gene, we tested the effects of camptothecin and of the approved TOP1 inhibitors Topotecan and SN-38 on TNFα-induced gene expression. RNA-seq experiments showed that inhibition of TOP1 but not of TOP2 activity suppressed the vast majority of TNFα-triggered genes. The TOP1 effects were fully reversible and preferentially affected long genes. TNFα stimulation led to inducible recruitment of TOP1 to the gene body of *IL8*, where its inhibition by camptothecin reduced transcription elongation and also led to altered histone H3 acetylation. Together, these data show that TOP1 inhibitors potently suppress expression of proinflammatory cytokines, a feature that may contribute to the increased infection risk occurring in tumor patients treated with these agents. On the other hand, TOP1 inhibitors could also be considered as a therapeutic option in order to interfere with exaggerated cytokine expression seen in several inflammatory diseases.

## 1. Introduction

Transcription is controlled by chromatin regulation that involves direct modification of DNA and histones, but increasing evidence suggests that also the topological properties of the DNA itself have the capacity to affect mRNA expression [1]. According to the “twin supercoiled domain model”, positive supercoils occur in front of the advancing RNA polymerase II (RNAPII) and negative supercoils trail the enzyme [2]. The resulting torsional stress in the DNA can be relieved by topoisomerase I (TOP1) which generates transient single-strand breaks, topoisomerase II (TOP2) that induces transient double-strand breaks or further enzymes such as APOBEC3B (apolipoprotein B mRNA editing enzyme catalytic subunit 3B) [3,4,5]. TOP1 and TOP2 have well documented roles in DNA replication, but increasing evidence suggest also their importance for efficient transcription [6,7,8]. TOP1 is a type IB enzyme that can relax both negative and positive supercoils by transient single-strand breaks using a process involving the covalent attachment of TOP1 to the double stranded DNA called topoisomerase cleavage complex (TOP1cc). One DNA strand is then cleaved, allowing the helical duplex to freely rotate around the unbroken strand followed by re-joining of the phosphodiester backbone of the DNA via the ligase activity of TOP1 [9]. TOP1 inhibitors such as camptothecin (CPT) inhibit re-ligation and stabilize the TOP1cc by intercalating at the TOP1-DNA interface [9]. CPT and its clinically used derivatives topotecan (TPT) or SN-38 are used for treatment of colon cancer, gynecologic malignancies and small-cell lung cancers [10,11], but also serve as tools to study the role of TOP1 in gene expression. TOP1 plays an important role in the regulation of basal and inducible gene expression by a variety of mechanisms. The enzyme was found at gene promoters where it participates in the release of the paused RNAPII by BRD4 (bromodomain containing protein 4)-dependent phosphorylation of RNAPII which stimulates TOP1 activity and thereby transcription elongation [12]. TOP1 was also found to be relevant for transcription elongation [13], suggesting that TOP1 can positively affect transcription at multiple levels. Overrepresentation of TOP1 has also been observed at super-enhancers where TOP1 can bind to transcription factors such as AIRE [14] and contributes to the synthesis of enhancer RNA (eRNA) at androgen-receptor-regulated enhancers [15].

Inducible gene expression is not only triggered by steroid hormones, but also by inflammatory processes where cytokines such as tumor necrosis factor α (TNFα) elicit a profound transcriptional response to trigger the expression of further inflammatory mediators such as interleukin 8 (*IL8*) and C-X-C Motif Chemokine Ligand 2 (*CXCL2*) in order to amplify the immune response [16,17]. Binding of TNFα to TNF receptor 1 enables the assembly multi-protein complexes which finally lead to the activation of the inducible transcription factor nuclear factor-κB (NF-κB), a complex process that depends on inducible degradation of the inhibitory IκBα protein [18]. NF-κB activity is subsequently terminated by a variety of autoregulatory feedback loops that involve the re-synthesis of IκBα proteins [19]. Inducible expression of inflammatory cytokines by NF-κB depends on the strongly transactivating p65 subunit, which harbors two transactivation domains (TADs) in its C-terminal region [20]. These TADs contact components of the basal transcription machinery and translate p65-DNA binding into productive transcription executed by RNAPII [21,22,23]. Subsequently, the C-terminal domain (CTD) of the largest RNAPII subunit is dynamically phosphorylated to coordinate the recruitment of factors allowing transcription initiation and elongation [24]. Following transcription and translation, cytokines such as interleukin 1 (IL-1) and TNFα are secreted from cells to coordinate both, locally and systemically the inflammatory host cell response to tissue damage and infection [25]. However, dysregulation of cytokine expression including a slight but permanent elevation of cytokine production can lead to chronic smoldering inflammation, while exaggerated and overshooting inflammation as it occurs for example in sepsis results in a "cytokine storm" that may result in tissue damage and even organ failure [17,26,27,28].

Here we investigated the consequences of TOP1 and TOP2 inhibition for TNFα and IL-1-induced gene expression. The use of various and also clinically used TOP1 inhibitors strongly interfered with a large part of TNFα-triggered gene expression in a fully reversible fashion. TOP1 inhibition preferentially inhibited long proinflammatory genes. As exemplified by the TNFα-induced and TOP1 sensitive *IL8* gene, TNFα leads to the recruitment of TOP1 to the gene body, thus facilitating transcription elongation. The clinical implications for cancer patients treated with TOP1 inhibitors and for patients suffering from exaggerated cytokine production are discussed.

## 2. Results

### 2.1. Effects of Various Clinically Used TOP1 and TOP2 Inhibitors on TNFα-Triggered Gene Expression

To test a potential contribution of TOP1 or TOP2 on TNFα-induced expression of inflammatory genes, we measured the impact of specific TOP1 inhibitors on TNFα-triggered gene expression in human diploid colon cancer HCT116 cells. Incubation of cells with the TOP1-selective inhibitor CPT [29] resulted in a strong and dose-dependent inhibition of inducible *IL8* expression, while the inhibitory effects on *CXCL2* transcription remained moderate (Figure 1A). In contrast, interference with TOP2 activity by ICRF193 did not affect TNFα-triggered expression of these two genes (Figure 1B).

Control experiments ensured that the inhibitory effect of CPT was not attributable to reduced cell viability in HCT116 and KB cells (Appendix A). It was then interesting to test whether also further approved TOP1 and TOP2 inhibitors display similar effects. Administration of TPT or SN-38, a biological active metabolite of irinotecan [30,31], strongly interfered with the TNFα-induced expression of *IL8*, *NFKBIA* (NF-κB inhibitor α), *TNFAIP3* (TNFα Induced Protein 3) and *ICAM1* (intercellular adhesion molecule 1), while inhibition of *CXCL2* and *CXCL10* expression was less pronounced (Figure 1C). Preincubation of cells with the TOP2 inhibitors teniposide or etoposide failed to interfere with TNFα-triggered expression of *IL8*, *NFKBIA, TNFAIP3, CXCL2*, *CXCL10* or *ICAM1* (Figure 1D), thus revealing that the observed effects are not restricted to one specific inhibitor. To investigate the effects of TOP inhibitors on untransformed cells we used conditionally immortalized human foreskin FS4-LTM fibroblasts that only proliferate in the presence of doxycycline. Also the TNFα-triggered gene expression in these FS4-LTM fibroblasts was efficiently inhibited by TOP1 inhibitors (Figure 1E). The effect of CPT on inducible gene expression was also seen at the protein level. HCT116 cells showed rapid IκBα phosphorylation and degradation upon short-term exposure to TNFα, followed by re-synthesis of IκBα after 60 min (Figure 1F). This re-synthesis of IκBα was completely absent in the presence of CPT. Also upstream signaling events were mildly affected by CPT, as detected by a reduction of TNFα-induced p65 Serine 468 phosphorylation in the presence of this TOP1 inhibitor (Figure 1F).

### 2.2. A General and Supportive Role of TOP1 for the Induction of the TNFα-Triggered Gene Response

So far, the experiments revealed a gene-specific effect of TOP1 inhibitors on TNFα-triggered gene expression. This gene specificity might be due to various reasons including the differential involvement of distinct pro-inflammatory transcription factors such as NF-κB or activator protein 1 (AP1), which cooperate to trigger expression of inflammatory genes [32,33]. In order to investigate the relative contribution of the strongly transactivating NF-κB p65 subunit on TNFα-triggered gene expression and its modulation by CPT, we generated p65-deficient HCT116 cells using the CRISPR-Cas9 system (Appendix A). Two single-cell clones lacking p65 expression showed defect TNFα-triggered expression of *IL8*, *CXCL2* and *NFKBIA* (Appendix A) and also no IκBα re-synthesis following its TNFα-induced proteasomal degradation (Appendix A), showing the successful elimination of NF-κB function. The single-cell clone 32 lacking p65 and also HCT116 wildtype (WT) cells were used to determine the effects of CPT and ICRF193 on basal and TNFα-induced gene expression by RNA-seq. Treatment with TNFα resulted in the induction of well-known target genes [34] and accordingly a gene ontology (GO) analysis revealed the involvement of these target genes in biological processes associated with infection and immunity (Appendix A).

TNFα-triggered gene expression was strongly reduced in the presence of CPT, while ICRF193 had no inhibitory effect in HCT116 WT cells. NF-κB p65-deficient cells showed almost complete abolishment of TNFα-inducible gene expression and thus did not allow to measure the effects of CPT and ICRF193, as visualized in a box plot (Figure 2A) and heat map (Figure 2B). Therefore, the subsequent bioinformatic analysis was only conducted for the analysis of HCT116 WT cells. A PCA of all genes revealed no effect of ICRF193 on basal gene expression in the absence of TNFα stimulation, while CPT already caused changes in basal and also in TNFα-induced transcription (Figure 2C). A PCA analysis of all TNFα-regulated genes showed the general inhibition of most TNFα-regulated genes by CPT (Figure 2A–C), while it exerted both activating as well as inhibitory functions on basal gene expression (Figure 2D). A GO analysis of the biological processes exerted by CPT-regulated genes revealed the involvement of upregulated genes in sensory perception and receptor signaling, while downregulated genes often participate in the (epigenetic) regulation of gene expression (Appendix A). 42 genes showed >2-fold upregulation by TNFα, but only 40 of them were found under all conditions (listed in Appendix A). In the presence of CPT only 6 of these 40 genes were still induced >2-fold, showing that only a minor fraction of the TNFα-induced mRNAs escapes from the inhibitory effect of CPT, as visualized in the Venn diagram displayed in Figure 2E. In contrast, 24 of the 40 genes were still upregulated >2-fold in the presence of ICRF193, reinforcing the notion that TOP2 inhibitors have only marginal effects on inflammatory gene expression. Previous studies suggested that especially long genes critically rely on the enzymatic activity of TOP1 [35,36,37]. Indeed, the analysis of TNFα-stimulated cells showed that the inhibitory effect of CPT had a tendency to be more pronounced in long genes (Figure 2F).

In contrast, the minor effects of ICRF193 on the stimulatory effects of TNFα showed no correlation with gene length (Figure 2F). Interestingly, the correlation with gene length was only seen for TNFα-regulated genes, while this correlation was statistically below significance for the effects of CPT and ICRF193 on basal gene expression (Figure 2G).

### 2.3. The Positive Effect of TOP1 on Gene Expression Does Not Strictly Correlate with the Magnitude of Gene Induction

The inhibitory effect of CPT in HCT116 cells was more pronounced for strongly induced genes (Figure 3A), raising the possibility that increased supercoiling and torsional stress in genes undergoing strong upregulation may result in an increased need for TOP1 activity.

To address this possibility in a more systematic fashion, we employed an experimental system allowing scalable gene expression. Towards this goal we used different fusion proteins where the DNA-binding domain of the yeast transcription factor Gal4 was fused to different parts of the transactivating C-terminus of the NF-κB p65 protein (Figure 3B). These fusion proteins are characterized by different transactivation capacities [20] and were expressed in a HEK293 cell line containing a stably integrated luciferase reporter gene driven by five Gal4 binding sites [38], correct expression was ensured via western blot (Figure 3C). In this experimental system CPT reduced luciferase gene expression about 50% and this effect occurred to a comparable extent irrespective of the strength of gene induction (Figure 3D). Again, TOP2 inhibition had only mild effects on reporter gene expression (Figure 3D). These data suggest that the relative strength of gene expression per se is not sufficient to explain the gene-specific effects of CPT, which may rely on additional parameters such as gene length, chromatin state or further factors.

### 2.4. IL-1-Triggered Gene Expression Is Also Supported by TOP1

Can TOP1 inhibition also interfere with inducible gene expression triggered by further cytokines such as IL-1? To address this question, we used human epithelial KB cells, which respond well to IL-1 [39]. KB cells were pre-treated with CPT and then stimulated either with TNFα or with IL-1, followed by the quantitative analysis of the inflammatory target genes *IL8*, *CXCL2* or *NFKBIA* using RT-qPCR (Figure 4A). CPT interfered with the expression of all these cytokine-induced genes irrespective of the inducing stimulus, though to a lesser extent as in HCT116 cells (Figure 1A–D). Accordingly, the inhibiting function of CPT is not restricted to one specific cytokine. Additionally, the clinically relevant TOP1 inhibitors TPT and SN-38 interfered with IL-1-induced expression of *NFKBIA* and *TNFAIP3*, but remained without impact on *CXCL10* expression (Figure 4B), similar to the observations in HCT116 cells (Figure 1C). In contrast, the TOP2 inhibitors ICRF193, amsacrine and etoposide showed no inhibitory function on IL-1-triggered gene expression in KB cells (Figure 4C), reinforcing the notion that TOP2 is without significant relevance for inflammatory gene expression, regardless of stimulus or cell type. Control experiments ensured that the inhibitory effect of CPT was not attributable to reduced cell viability (Appendix A).

### 2.5. The Effect of TOP1 on Inflammatory Gene Expression Is Fully Reversible and Preferentially Affects Early Induced Genes

It was then interesting to determine the kinetics of reversible TOP1 inhibition using an experimental set-up that is schematically displayed in the upper part of Figure 5A.

HCT116 WT cells were incubated with CPT or vehicle for 2 h prior to TNFα stimulation, followed by washing of cells and further incubation in normal medium for various periods. Subsequently the cells were stimulated for 1 h with TNFα, followed by the analysis of several early induced genes by RT-qPCR. The inhibitory effect of CPT on gene induction was lost in a time-dependent manner and already strongly diminished at 4 h after washout of the inhibitor *(IL8*, *CXCL2* and *NFKBIA*), or even completely vanished 2 h after CPT removal (*TNFAIP3*) (Figure 5A). These data points are in good agreement with studies determining the clinical pharmacokinetics of TPT showing a mean elimination half-life of ~3 h [40].

The inflammatory gene response occurs in a timely coordinated manner with genes expressed at early, intermediate and late time points [41]. To test whether CPT preferentially affects early or late genes, HCT116 cells were stimulated for 1 or 8 h with TNFα in the absence or presence of DMSO, CPT or ICRF193, followed by the analysis of gene expression by RT-qPCR. For the analysis of TOP effects on early induced genes, cells were pre-treated for 2 h with the respective TOP inhibitors, prior to additional TNFα-treatment for 1 h. For the analysis of late induced genes which might also require the de novo synthesis of transcriptional regulators, cells were stimulated for 5 h with TNFα, followed by the intermittent, additional administration of TOP inhibitors and further incubation, as schematically shown on the top of Figure 5B. These analyses showed that genes having their maximal expression after 1 h of TNFα (type 1 genes) and genes having comparable levels of gene expression after 1 or 8 h (type 2 genes) showed a strong inhibitory effect by CPT after 1 h, but no inhibition at the late time points (Figure 5B). In contrast, the type 3 genes represented by *CXCL10* and *Ccl5* reached their maximal expression after 8 h and showed no statistically relevant CPT-mediated inhibition of gene expression (Figure 5B). These experiments raise the possibility that CPT preferentially affects expression of early induced genes, but general conclusions on this require the systematic analysis of many genes in various cell types on a genome-wide level.

### 2.6. CPT Interferes with TNFα-Induced Histone H3 Acetylation and Transcription Elongation

The inhibitory effects of TOP1 inhibitors on gene expression may rely on several mechanisms, such as impaired DNA-binding of NF-κB p65, RNAPII or on defects in transcription initiation and elongation. To distinguish between these possibilities, we performed chromatin immuno-precipitation (ChIP) experiments covering various upstream and transcribed regions of the well characterized and CPT-sensitive *IL8* gene, as schematically displayed in Figure 6A. HCT116 cells were stimulated for 1 h with TNFα in the absence or presence of CPT and subjected to ChIP using antibodies recognizing p65, RNAPII, elongating (phospho-serine 2 CTD) and initiating (phospho-Serine 5 CTD) RNAPII as well as TOP1. TNFα stimulation caused the association of p65 to its binding sites at the enhancer and the promoter of the *IL8* gene in a CPT-independent fashion (Figure 6B). In support to this result, fractionation experiments showed that TOP1 and TOP2 inhibitors had no impact on the TNFα-induced translocation of NF-κB p65 to the nuclear and the chromatin fraction (Appendix A). While TNFα caused the recruitment of RNAPII to the promoter and the downstream transcribed regions in exons 1 and 2, the inhibition of TOP1 resulted in a slight inhibition of RNAPII occupancy at these sites. TNFα increased the amount of initiating RNAPII at the promoter and the downstream transcribed introns and exons in a CPT-independent fashion. In contrast, CPT prohibited the TNFα-induced occurrence of elongating RNAPII at the transcribed regions of the *IL8* gene (Figure 6B), suggesting that the inhibitory effect of CPT on inflammatory gene expression also depends on interference with transcription elongation rather than transcription initiation. Intriguingly, TNFα stimulation also led to a slight but consistently observed increase of TOP1 association with the transcribed region at exons 1 and 2 and intron 1. This TOP1 association was less pronounced in the presence of CPT (Figure 6B). TOP1 participates in re-shaping the chromatin structure of its target genes and shows enrichment at chromatin marks such as H3K4me3 [42,43]. As inflammatory cytokines subsequently also lead to changes in histone modifications at inflammatory genes including *IL8* [39], it was interesting to test whether CPT might also affect these events. HCT116 cells were treated for 1 h with TNFα in the absence or presence of CPT, followed by ChIP experiments using antibodies recognizing the active enhancer mark H3K27ac [44], and H3K9ac, a modification indicating the switch from transcription initiation to elongation on promoters [45]. TNFα caused the enrichment of H3K27ac and H3K9ac at the enhancer-flanking regions while H3K9ac was also induced at exons 1 and 2 and intron 1. All TNFα-induced H3 acetylations were impaired in the presence of CPT (Figure 6C), suggesting that TOP1 inhibitors either also affect chromatin modifications or *vice versa* that changes in gene expression also affect chromatin modifications.

## 3. Discussion

Here we show that several TOP1 inhibitors interfere with cytokine-induced and NF-κB p65-mediated inflammatory gene expression in a variety of cell lines. This effect -in combination with a previously reported loss of white blood cells due to hematological toxicity- may contribute to the increased risk of infections occurring in cancer patients treated with TOP1 inhibitors [46]. Vice versa, the reported inhibitory activity of TOP1 inhibitors on the LPS- and virus-induced host response together with its protective effect in a model of LPS-induced cell death repurposed TOP1 inhibitors as agents for the treatment of septic shock [47]. Recent evidence also showed an alleviation of LPS-mediated acute lung injury by TPT [48]. Conditions associated with systemic release of large amounts of cytokines or even a cytokine storm such systemic inflammatory response syndrome (SIRS), sepsis or graft-versus-host disease are difficult to treat and can lead to the cytokine-induced lethal multiple organ dysfunction syndrome [49]. The potent inhibitory effect of clinically used TOP1 inhibitors on inflammatory gene expression and its full reversibility suggest that TOP1 inhibitors might be therapeutically used for a limited period to antagonize excessive and damaging cytokine production.

The induction of single-strand breaks by CPT has been reported to activate NF-κB, but interestingly a robust induction of DNA-binding activity by CPT and other DNA-damaging agents is typically contrasted by minor changes in the expression of endogenous NF-κB target genes [50,51,52]. Accordingly, we found that CPT treatment of cells induced the expression of only two NF-κB target genes (*LYRM5* and *C18orf32*), but both genes were only induced with a LFC <1.

Genetic studies in yeast revealed the strong requirement of TOP1 and TOP2 for the activation of genes characterized by high transcriptional plasticity [53]. However, some genes only require the function of one class of topoisomerases. This is exemplified by the transcription of neuronal early-response genes and hormone-induced estrogen receptor-α (ERα) target genes, which depend on TOP2β cleavage activity [8,54]. In contrast, this study shows that cytokine-induced inflammatory gene expression relies exclusively on the activity of TOP1, while TOP2 activity was fully dispensable for gene activation. The molecular mechanisms responsible for this bias are currently not clear. Of note, TPT also induces DNA double-strand breaks at sites distinct from Top1cc covalent complexes [55,56], thus we can formally not rule out a role of double-strand DNA breaks in inflammatory gene expression.

Another interesting implication of our results is the fact that TOP1 inhibition affects the large majority of TNFα-induced genes. This could be due to several reasons including the fact that many cytokines are clustered on genomic regions as exemplified by the prototypical CXCL-chemokine locus on human chromosome 4 [57]. In this case, TOP1-mediated relief of positive and a negative DNA supercoiling would affect also neighbouring cytokine genes. In support to this concept, genome-wide chemical probing of the DNA structure has revealed the existence of DNA supercoiling domains characterized by an enrichment of TOP1, accessible chromatin and a depletion of TOP2 [58]. Another possible mechanism relies on the ability of TOP1 to interact with specific DNA-binding proteins such as AIRE or NKX3.1 and with chromatin remodelers including SMARCA4 and BAZ1B-SMARCA5 [14,43,59,60]. In this case, association of TOP1 with master transcription factors of the inflammatory gene response such as NF-κB or AP1 would blunt a large part of proinflammatory gene expression program. Furthermore, TOP1 can interact with the AP1 subunit c-Jun [61] and it will be interesting in future studies to systematically investigate the interactomes and posttranslational modifications of TOP1 proteins in cytokine-stimulated cells. These experiments could also clarify the molecular mechanisms allowing for TNFα-inducible recruitment of TOP1 to the bodies of inflammatory genes.

TOP1 inhibition resulted in the diminished occurrence of elongating RNAPII in the transcribed region of the *IL8* gene. Given the general role of TOP1 for transcription elongation revealed by sequencing of nascent RNA [13], we speculate that diminished transcription elongation also accounts for impaired transcription of most other proinflammatory genes. The atypical kinase BRD4 does not only trigger Serine 2 phosphorylation in the elongating RNAPII CTD, but also stimulates TOP1 activity to overcome torsional stress [12]. In addition, a stabilized TOP1-DNA complex can impose a physical obstacle for elongating RNAPII, thereby hindering its progression through the gene body [62,63]. But also, other components can account for the importance of TOP1 activity, as its inhibition changed H3 acetylation at the *IL8* locus and most probably will also cause changes in chromatin compaction [64] or the expression of non-coding RNAs which are known to assemble TOP1 containing protein complexes [15,65,66].

## 4. Materials and Methods

### 4.1. Cell Culture and Transfections 

The human cancer cell lines KB, human embryonic kidney (HEK)293 and diploid HCT116 colon cancer cells were maintained in DMEM GlutaMAX (TM) (Thermo Fisher Scientific, Waltham, MA, USA) supplemented with 10% fetal calf serum and 100 U/mL penicillin and 100 µg/mL streptomycin at 37 °C in a humidified atmosphere containing 5% CO_2_. NF-κB p65-deficient HCT116 cells were generated via CRISPR-Cas9 by transfection of 6 µg of the pX459 vector containing a single guide RNA targeting the third exon of p65 into 2000 cells per 10-cm dish using linear polyethylenimine [67]. After one day the non-transfected cells were eliminated by the addition of puromycin (1 µg/mL). After 48 h the dead cells were aspirated off and DMEM GlutaMAX without puromycin was added to allow the growth of single cell-derived clones. Single clones were picked and further analyzed for expression of Cas9 and p65, the genomic indel mutations were characterized by PCR amplification and sequencing. Primary human FS4-LTM fibroblasts were grown in huFIB Medium (InSCREENeX GmbH, Braunschweig, Germany) and their proliferation was induced by addition of (1 µg/mL) doxycycline.

### 4.2. Cell Extraction and Western Blotting

Cells were washed in PBS and collected by centrifugation. The cell pellet was resuspended in RIPA lysis buffer (10 mM Tris/HCl, pH 7.5; 150 mM NaCl; 0.5 mM EDTA; 0.1% SDS; 1% Triton X-100; 1% deoxycholate) freshly supplemented with 10 mM NaF, 0.5 mM sodium orthovanadate, 1 mM phenylmethylsulfonylfluoride, leupeptin (5 µg/mL) and aprotinin (10 µg/mL). Following incubation on ice for 30 min, the extracts were centrifuged and equal amounts of proteins contained in the supernatant were further analyzed. Western blotting was performed by SDS-PAGE and semi-dry transfer to polyvinylidene difluoride membranes. After incubation with primary and secondary peroxidase-coupled antibodies, proteins were detected using Western-Lightning Plus-ECL solutions (PerkinElmer, Waltham, MA, USA) according to the instructions given by the manufacturer. All bands were visualized using the ChemiDocTM XRS+ System (Bio-Rad Laboratories, Munich, Germany). Relative intensity ratios were calculated using the Bio-Rad Image Lab-software.

### 4.3. Quantification of Gene Expression by RT-qPCR

Cell were left untreated or pre-treated with TOP1 and TOP2 inhibitors or vehicle, the concentrations, incubation times and inhibitors are specified in the figure legends. Cell stimulation was done upon addition of 20 ng/mL of TNFα or 10 ng/mL of IL-1. Total RNA was isolated using the RNeasy kit (Qiagen, Hilden, Germany) according to the procedure described by the manufacturer. One microgram of RNA was used for the generation of cDNA using Oligo (dT) 12-18 primers and SuperScript II Reverse Transcriptase (Thermo Fisher Scientific). The cDNA was diluted and used for the RT-qPCR reactions using the SYBR green reporter dye. Every reaction was performed as duplicates and quantified with the ΔΔCT-method. Threshold cycles (CT) of target genes were normalized to a housekeeping gene (TPI). The resulting ΔCT were compared to control samples and relative mRNA expression was calculated by R = 2^−ΔΔCT^.

### 4.4. Luciferase Reporter Assays

HEK293 cells with a stably integrated firefly luciferase reporter gene driven by five Gal4 binding sites [38] were seeded in 6-well plates. The next day, cells were transfected with 1 µg of Gal4-constructs together with a plasmid encoding Renilla luciferase. After 24 h cells were treated with 10 µM of CPT, ICRF193 or vehicle and incubated for another 8 h. Cells extracts were prepared with the Dual Luciferase Assay Kit (Promega, Madison, WI, USA) and firefly and Renilla luciferase activity was measured in a luminometer (Duo Lumat LB 9507, Berthold, Bad Wildbad, Germany) as described [32]. All values were normalized to the activity of Renilla luciferase.

### 4.5. RNA-Seq and Bioinformatics Analysis 

HCT116 cells were left untreated or treated for 2 h with 5 µM CPT, ICRF193 or vehicle. 20 ng/mL of TNFα were additionally added for 1 h as specified in the figure legends. Total RNA was isolated using the NucleoSpin RNA Kit (Macherey-Nagel, Düren, Germany) and RNA quality was assessed using the Experion RNA StdSens Analysis Kit (BioRad). RNA-Seq libraries were prepared from total RNA using the TruSeq Stranded total RNA LT kit (Illumina, San Diego, CA, USA) and the further analysis was performed as published [32]. The statistical analysis of read counts is given in Appendix A, RNA-Seq fastq files were controlled for quality issues using fastqc (https://www.bioinformatics.babraham.ac.uk/projects/fastqc/). Trimming and filtering was done by trimmomatic (trimmomatic SE -threads 8 HEADCROP:10 LEADING:3 TRAILING:3 SLIDINGWINDOW:4:15 MINLEN:36) [68]. Gene counts were called by FeatureCounts [69]. Coverage profiles were created by Deeptools bamCoverage function using FPKM normalization [70]. Differential gene expression analysis was done with the R/Bioconductor package DESeq2 [71]. DESeq2 internal Principal Component Analysis (PCA) was used to show the similarity between different samples. Differentially expressed genes were chosen using the FDR *p*-value cut-off < 0.05 and an absolute log2 fold change > 1. The variability between replicates is modeled by the dispersion parameter of the R/Bioconductor package DESeq2. DESeq2 assumes that genes of similar average expression strength have similar dispersion. The R/Bioconductor package ggplot2 was used to plot the heatmaps and calculate the smooth curve with a linear model fitting. The R/Bioconductor package ggplot2 was used to plot the heatmaps [72] and the package VennDiagram to plot venn plots (https://CRAN.R-project.org/package=VennDiagram). The data generated in this study are available at the GEO repository with the accession number (GSE128798). GO analyses were done using the Bioconductor package clusterProfiler [73].

### 4.6. Chromatin-Immunoprecipitation (ChIP)

One day prior to the experiment 1x 107 HCT116 were seeded in T175 flasks. The next day, cells were treated for 2 h with 5 µM CPT or left untreated. TNFα (20 ng/mL) was added for 1 h and cells were cross-linked with formaldehyde (1% (*v*/*v*) final concentration) for 10 min, followed by addition of glycine (100 mM final concentration) for 5 min. Cells were collected in the medium using a cell scraper and immediately put on ice. Three flasks from the same condition were pooled. After centrifugation for 5 min at 4 °C and 3000 rpm, the supernatant was aspirated off and the pellet was resuspended in 2 mL ice cold PBS with PMSF (0.5 mM final concentration). Following centrifugation for 5 min at 4 °C and 3000 rpm, supernatant was aspirated off and cells were resuspended in ice cold lysis-buffer (1% (*w*/*v*) SDS, 10 mM EDTA, 50 mM Tris/HCl pH 8.1, 0.5 mM PMSF, cOmpleteTM Protease Inhibitor Cocktail (Roche, Basel, Switzerland)). Lysis took place on ice for 10 min. 1 mL of the lysate was transferred to milliTube 1 mL AFA Fiber tubes (Covaris, Woburn, MA, USA) and sonicated in a Covaris S220 device with the following program: peak incident power: 150 W; duty factor: 15; cycles per burst: 500 for 30 s followed by 2.5 W; duty factor: 15; cycles per burst: 500 for 30 s. This program was repeated 20 times and generated DNA fragments in a size ranging between ~150 and ~300 bp. Sonicated lysates were centrifuged for 15 min at 13,200 rpm at 4 °C and supernatants transferred to new reaction tubes. Aliquots representing 25 µg of chromatin were diluted 1:10 with dilution buffer (0.01% (*w*/*v*) SDS, 1.1% (*v*/*v*) Triton X-100, 1.2 mM EDTA, 167 mM NaCl, 16.7 mM Tris/HCl pH 8.1) and subjected to 4 h of preclearing with an agarose A/G-bead-mixture and 2 µg rabbit IgG antibody at 4 °C with end-over-end tumbling. Supernatants were incubated with the antibodies listed in Appendix A. Immunoprecipitation was carried out over night with end-over-end tumbling at 4 °C. A mixture of agarose A/G-beads was added for 4 h, followed by successive washing of immunoprecipitated complexes for 5 min at 4 °C with end-over-end tumbling using low-salt buffer (0.1% (*w*/*v*) SDS, 1% (*v*/*v*) Triton X-100, 2 mM EDTA, 150 mM NaCl, 20 mM Tris/HCl pH 8.1), high-salt buffer (0.1% (*w*/*v*) SDS, 1% (*v*/*v*) Triton X-100, 2 mM EDTA, 500 mM NaCl, 20 mM Tris/HCl pH 8.1), LiCl-buffer (250 mM LiCl, 1% (*v*/*v*) NP40, 1% (*w*/*v*) deoxycholate, 1 mM EDTA, 10 mM Tris/HCl pH 8.1), and twice with TE-buffer (10 mM Tris/HCl pH 8.1, 1 mM EDTA). Reverse-crosslinking took place in TE-buffer with addition of RNase A for 30 min and Proteinase K for 2 h at 37 °C followed by incubation at 65 °C overnight. Free DNA was purified using NucleoSpin Gel and PCR Clean Up Kit with buffer NTB (Macherey-Nagel) and was eluted in 50 µL Elution buffer. The amount of immunoprecipitated DNA was analyzed via RT-qPCR, calculation of enrichment by immunoprecipitation relative to the signals obtained for 1% (*v*/*v*) input DNA was performed. The primers used for the detection of the various genomic regions at the *IL8* locus were previously published [39] and are listed in Appendix A.

### 4.7. Antibodies, Plasmids, Oligonucleotides and Reagents

This information is given in Appendix A.

## 5. Conclusions

This study shows a major block of TNFα-induced inflammatory and NF-κB-dependent gene expression after inhibition of TOP1. Interference with the enzymatic activity of TOP1 at gene bodies impairs transcription elongation. These findings are of broad biomedical relevance and include cancer patients treated with TOP1 inhibitors.

## Figures and Tables

**Figure 1 cancers-11-00883-f001:**
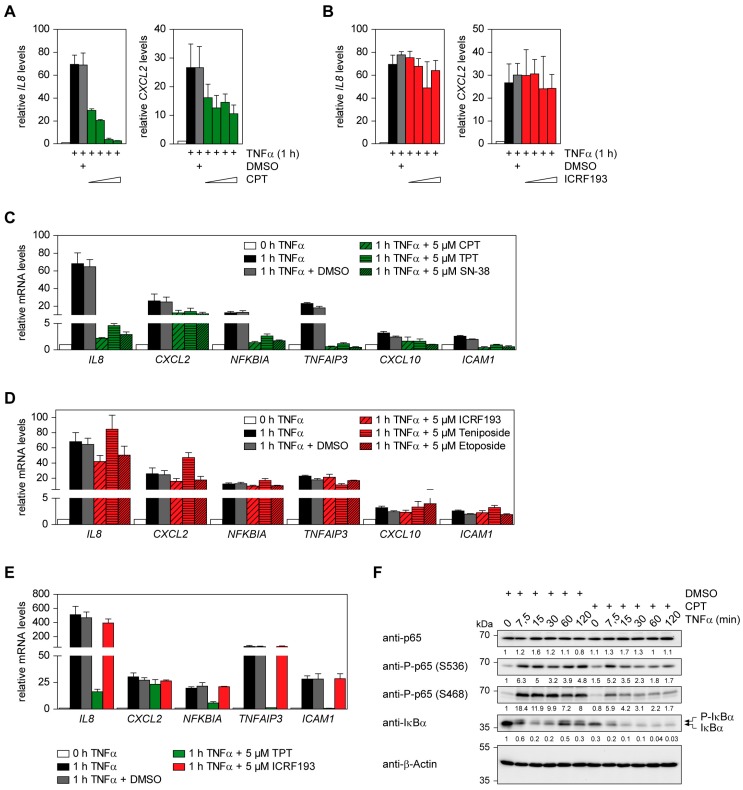
Effect of TOP1 and TOP2 inhibitors on TNFα-induced inflammatory gene expression in HCT116 and FS4-LTM cells. (**A**,**B**) HCT116 cells were pre-treated for 2 h with increasing (0.5 µM, 1 µM, 5 µM, 10 µM) concentrations of CPT (**A**) or ICRF193 (**B**) or vehicle (DMSO) in the controls and then additionally stimulated for 1 h with TNFα. Cells were subsequently analyzed for *IL8* and *CXCL2* gene expression by RT-qPCR, error bars show SEMs obtained from at least two independent experiments performed in duplicate. (**C**,**D**) HCT116 cells were pre-treated for 2 h with 5 µM of various TOP1- (**C**) or TOP2- (**D**) inhibitors as shown, followed by the addition of TNFα (20 ng/mL) for 1 h. Expression of various indicated inflammatory NF-κB target genes was assessed via RT-qPCR. Error bars show SEMs obtained from three independent experiments performed in duplicate. (**E**) Primary human FS4-LTM fibroblasts where treated and analyzed as described for HCT116 cells in (**C**,**D**). SEMs were obtained from three independent experiments performed in duplicate. (**F**) HCT116 cells were pre-treated for 2 h with 5 µM of CPT or DMSO, followed by the addition of TNFα (20 ng/mL) for various periods. Protein lysates were prepared and equal amounts of protein were analyzed by Western blotting for the occurrence or phosphorylation of the indicated proteins. The positions of molecular weight markers are indicated. Normalized intensity ratios are given for each band, the intensity of the DMSO-treated control was set as 1. β-Actin was used as housekeeping protein to ensure equal protein loading, one out of three experiments is shown, the full blots are shown in Appendix A.

**Figure 2 cancers-11-00883-f002:**
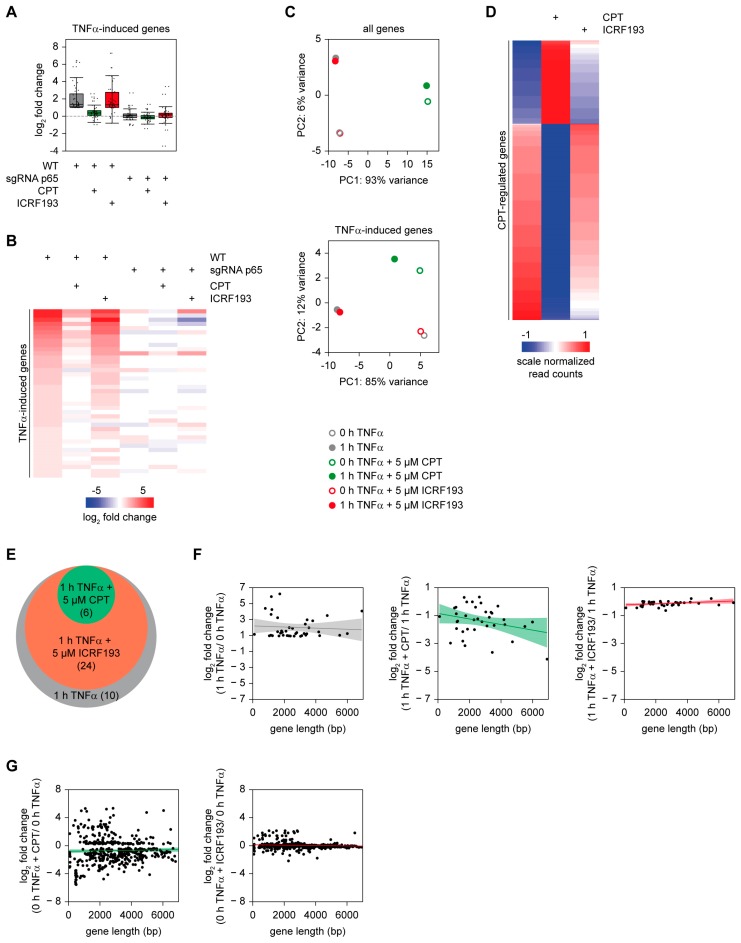
RNA-seq analysis of CPT effects on TNFα-triggered gene expression in HCT116 WT and p65-deficient HCT116 cells. (**A**) HCT116 and p65-deficient HCT116 cells were pre-treated for 2 h with CPT or ICRF193 (5 µM each), followed by stimulation for 1 h with TNFα (20 ng/mL). RNA-seq analysis results are depicted in a boxplot, each dot represents a regulated gene measured by ≥ 20 reads in at least one condition and undergoing an upregulation of > 1 log2 fold change (LFC) upon TNFα-stimulation. The boxed area displays the region containing the second and third quartile, the median is indicated. (**B**) Heat map visualization of the RNA-seq experiment, only genes with a LFC >1 and ≥ 20 reads in at least one condition are displayed (40 genes). (**C**) The upper part shows a two-dimensional PCA analysis of the RNA-Seq data for the indicated conditions in HCT116 WT cells. The lower part shows a PCA analysis for the 40 genes that are upregulated >2-fold in response to TNFα treatment. (**D**) Heat map visualization of CPT and ICRF193 effects on basal gene expression of HCT116 WT cells. As these TOP1 inhibitors can potentially affect housekeeping genes used for normalization, read counts are displayed. Gene counts are normalized by DESeq2 and filtered for significant (*p* value < 0.05) expressed CPT regulated genes (0 h TNFα + CPT versus 0 h TNFα (500 genes)). (**E**) Venn diagram visualizing the distribution of genes undergoing a >2-fold induction of TNFα-triggered gene expression in the absence or presence of the indicated inhibitors. (**F**) Correlation of the TNFα-induced changes in gene expression (in log2 scale) in the absence (left) or presence of inhibitors for TOP1 (middle) and TOP2 (right) versus the gene length. Only genes with a LFC >1 and ≥ 20 reads in at least one condition are displayed. The colored central line represents the linear smooth, the colored area displays the confidence interval around the smooth. (**G**) HCT116 WT cells were treated for 2 h with CPT or ICRF193 (5 µM each). Genes showing a LFC > 1 upregulation or downregulation and showing a *p*-value < 0.05 were not correlated with gene length.

**Figure 3 cancers-11-00883-f003:**
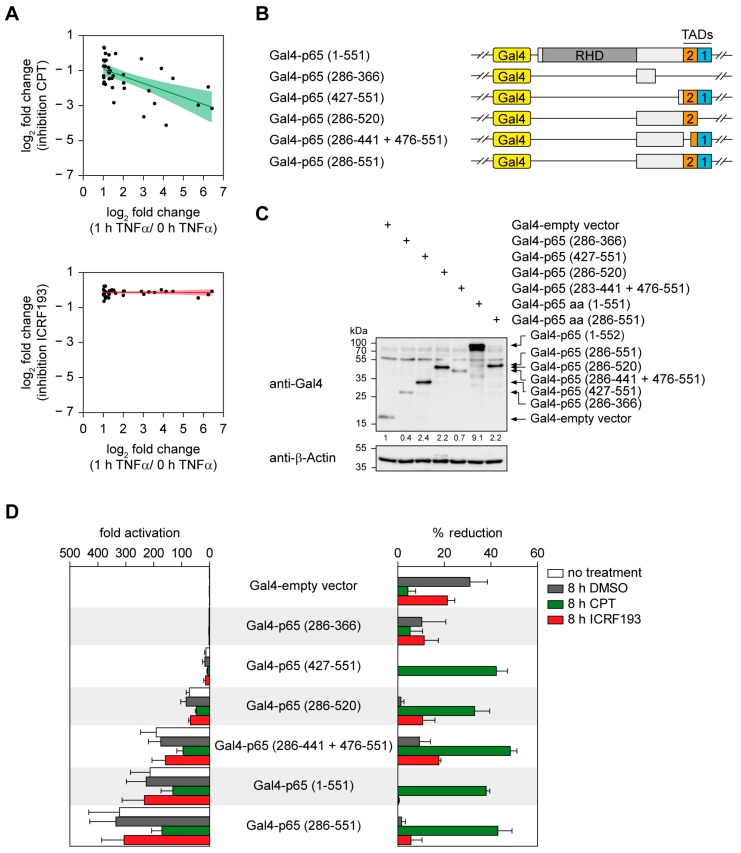
The contribution of TOP1 for gene expression is not correlated with the strength of gene induction. (**A**) Correlation of the strength of the inhibitory effects by CPT (upper; spearman correlation: −0.56, coefficients *p*-value: 2 × 10^−4^) or ICRF193 (lower; spearman correlation: −0.06, coefficients *p*-value: 0.8668) versus TNFα-induced changes in gene expression. All genes were selected for being TNFα-induced targets (LFC > 1 and ≥ 20 reads). The colored central line represents the linear smooth, the colored area displays the confidence interval around the smooth. (**B**) Schematic visualization of Gal4-p65 constructs containing either full-length p65 (aa 1-551) or truncated forms of p65. The position of the Rel homology domain (RHD) and of the transactivation domains (TADs) are indicated. (**C**) HEK293 cells with a stably integrated Gal4-driven firefly luciferase reporter gene were transfected with the various Gal4 constructs that are visualized in (**B**). Cellular extracts were used to confirm expression of the Gal4-p65 proteins by immunoblotting. The positions of molecular weight markers are indicated, normalized intensity ratios are given for each band. β-Actin was used as housekeeping protein, the whole blots are displayed in Appendix A. (**D**) The cells transfected as described in (**C**) were treated one day later for 8 h with 10 µM CPT, ICRF193 or DMSO or left untreated. Cell extracts were used to determine the luciferase activity which is displayed either as -fold activation (left) or alternatively as % reduction (right). Error bars show SEMs from three independent experiments.

**Figure 4 cancers-11-00883-f004:**
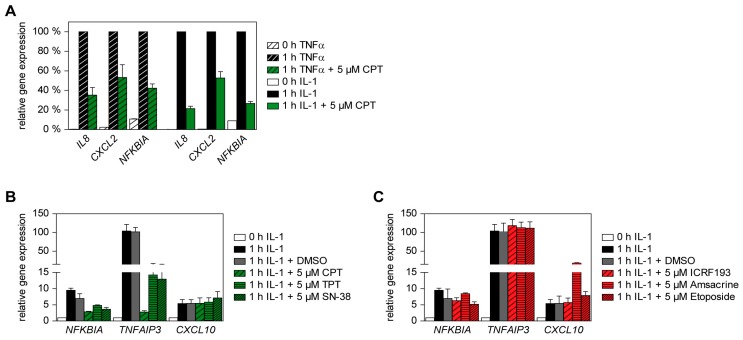
Effect of TOP1 inhibitors on IL-1-induced inflammatory gene expression in KB cells. (**A**) KB cells were pre-treated for 2 h with CPT and then stimulated for 1 h either with TNFα (20 ng/mL) or with IL-1 (10 ng/mL). Cells were harvested and analyzed by RT-qPCR for expression of *IL8*, *CXCL2* and *NFKBIA* as shown. To facilitate comparison, maximal gene expression for the respective genes was set as 100%, error bars display SEMs obtained from three independent experiments performed in duplicate. (**B**,**C**) KB cells were pre-treated for 2 h with 5 µM of various TOP1 (**B**) or TOP2 (**C**)-inhibitors as shown, followed by the addition of IL-1 (10 ng/mL) for 1 h. Expression of various indicated inflammatory NF-κB target genes was assessed via RT-qPCR. Error bars show SEMs obtained from three independent experiments performed in duplicate.

**Figure 5 cancers-11-00883-f005:**
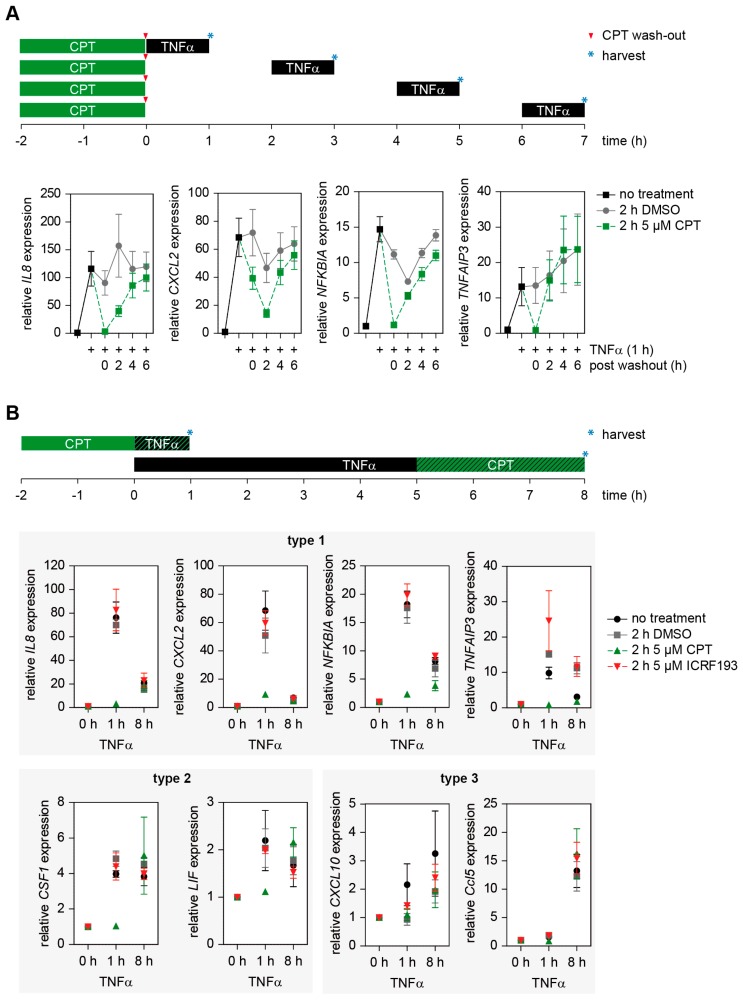
Reversibility and kinetic analysis of the CPT effect on TNFα-triggered gene expression. (**A**) Upper part: schematic of experimental design. HCT116 cells were treated for 2 h with 5 µM CPT or DMSO and cells were either directly stimulated with TNFα for 1 h or alternatively cells were washed with PBS and the medium was replaced by inhibitor-free medium. After the indicated time periods post-treatment, cells were stimulated for 1 h with TNFα. The lower part shows the quantification of mRNA expression levels of the indicated inflammatory genes using RT-qPCR. Error bars show the SEMs from five individual experiments. (**B**) HCT116 cells were either treated with TNFα for 5 h and additionally with 5 µM CPT, ICRF193 or DMSO for 3 h (8 h TNFα; 3 h CPT/ICRF193/DMSO treatment in total) or were treated for 2 h with CPT, ICRF193 or DMSO and additionally for 0 h and 1 h with TNFα (3 h of CPT/ICRF193/DMSO treatment in total). Gene expression levels were analyzed by RT-qPCR, error bars show SEMs from five individual experiments. Target genes were grouped according their kinetic behavior (group 1: maximal expression early, group 2: comparable expression early and late, group 3: maximal expression late).

**Figure 6 cancers-11-00883-f006:**
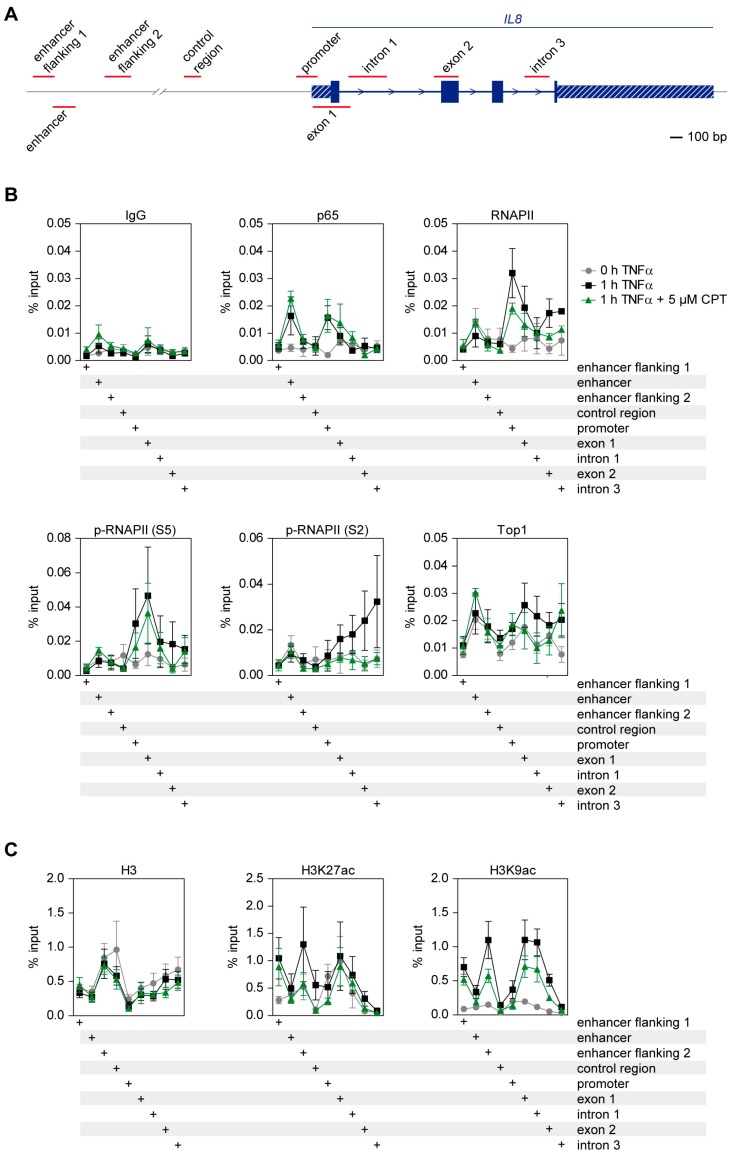
Effects of TOP1 inhibitors on transcription elongation, association of regulatory proteins and histone modifications. (**A**) Schematic display of the analyzed region covering parts of the *IL8* locus. Exons are shown by the filled boxed areas, 3′ and 5′ UTR are indicated by shaded boxes and the positions of the regions detected by PCR are shown in red. (**B**) HCT116 cells were either treated for 2 h with 5 µM CPT and additionally for 1 h with TNFα or were treated for 0 or 1 h with TNFα alone. Cells were subsequently subjected to ChIP analysis using the indicated antibodies, chromatin binding of the various proteins was quantified via qPCR. Error bars show SEMs from three individual experiments. (**C**) The experiments were done as in (**B**) with the difference that the levels of H3K27ac and H3K9ac-bound DNA were determined.

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
