# Peer review of "Chemotherapeutic Drugs Inhibiting Topoisomerase 1 Activity Impede Cytokine-Induced and NF-κB p65-Regulated Gene Expression"

_cancers, 2019, doi:10.3390/cancers11060883_

Round 1

Reviewer 1 Report

The authors report on the effects produced by Top1 inhibitors on the transcription of NF-kB regulated cytokine genes upon TNF stimulation.

They find that Top1 –but not Top2– inhibitors decreased the expression of TNF-induced genes. They also mention that the effect is reversible, and early and long genes are more affected than the others.

Results are interesting although similar results on Top1 inhibition on TLR activation has been recently published (see ref 51) and suggest a protective role for Top1 inhibitors in sepsis and exacerbated inflammation.

General comments:

Generalisations are the main defect of this paper (see comments for Fig 5 and 6). Discussion is the most affected part. Conclusions seem correct but must be implemented with more careful analyses.

The title itself puts excessive emphasis on a general conclusion that is not supported by data: results support the effect of Top1 inhibitors on p65-mediated TNF-induced transcription but this effect cannot be generalized to all the cytokines (only IL 1 appears in Fig 4) and to all the NF-kB transcription factors (only p65 mutants are considered) as indicated in the title that should be changed.

In my opinion, the authors can simply propose a relationship between CPT activity and p65-driven transcription provided that RNA seq analyses support such conclusion.

Statistics and replicates analyses are missing in most of the figures and poorly described in Mat&Meth.

The description of experiments and methods, as well as cartoons, must be simplified and/or more details must be provided (i.e. expmts in Fig5)

Few typos are present and should be corrected, like in lines, 172, 81…

Specific comments:

Figure 1:

CTP and TPT concentrations used are high as compared to what reported in literature and might be toxic for the cells. A viability assay is strongly suggested.

Panel E: how many times the experiment has been repeated? A quantification of WB plus statistics would be informative.

Figure2: Technical details like library size are missing for the RNA seq. How many replicates have been analysed for each experimental point? It seems just one. How can the FDR be calculated with one replicate? For the sake of reproducibility, at least one replicate/point should be added.

LFC is >1 in the caption but >0.5 in M&M. Which one has been used? The authors consider variation in transcription for only 42 genes. What about all the others? The other Differentially Expressed Genes, either up or down-regulated, are beneficial or detrimental? Gene Ontology/GSEA analyses should increase our understanding of the molecular mechanisms underlying TOP1 role in TNFregulated transcription. Can conclusions be drawn based on 42 genes only? By the way, table S2 shows only 40 genes and not 42... The authors should elaborate on these issues and show genomewide transcription.

Figure 3: again, conclusions about the correlation between CPT effects and transcription levels is drawn on 42 DEGs. Genomewide correlation analysis should be shown. Panel D: experimental replicates? Statistical analyses? SD are huge!

Figure 4: concerns on cell viability as for Figure 1.

Figure 5 suggests that kinetics and reversibility of CPT effects on early, intermediate and late genes is different, but only 1 or 2 genes per group were analysed. Variations have a very big standard deviation. Can the authors really draw such conclusions?

Moreover, the cartoon in (A) is misleading since it seems to describe pulses of TNF after CPT washout. Overall, the description of the expmt must be improved while the choice of either pretreating or cotreating the cells with CPT and/or TNF should be justified. In any case, I doubt that the results can be plot together. Please elaborate on this problem.

Figure 6 supports the idea that CPT is most effective on early and long genes but only IL8 gene transcription has been analysed so far. Generalization is not supported by data.

The scale bar and amplicons size are missing in the cartoon. Technical concern: exon 1 is 200bp long. ChIP is performed on sonicated chromatin with 500-600 bp size (assumed, but not shown/described). Therefore, it is impossible to distinguish the promoter from the exon1 region in the immunoprecipitated DNA by PCR as shown in panels B and C. The authors should explain and possibly correct the experimental design.

Author Response

Please find enclosed the details in the attached file including figures.

Reviewer 2 Report

In this manuscript Riedlinger et al describe a role of topoisomerase 1 (TOP1) inhibitors (CPT+ derivatives) in dampening TNF/IL1-induced NF-kB-dependent cytokine and pro-inflammatory gene expression in cancer cells. Overall, the data are interesting, however, there are some concerns that need to be addressed.

1.     There are several published papers that show that CPT increases TNF production and activates NF-kB in cancer cells, leading to apoptosis. The authors fail to mention these reports. In the current study, TNF and IL1 data are normalized to untreated cells w/o CPT treatment, not allowing to see whether this was also the case here. The authors need to show how CPT treatment alone affects NF-kB-dependent gene expression in this study, as effects on basal gene expression may have impact on induction capability.

2.     Supportive of such a pre-activation, the Western Blot in Figure 1 suggests lower IkBa protein levels and increased S536 p65 phosphorylation in CPT cells at timepoint 0, compared to non-treated cells. The authors claim no change in regulatory p65 phosphorylation with CPT in the text, which is not what the blot indicates.

3.     p65 nuclear translocation may be used as a reliable indicator of NF-kB activation in addition to phosphorylation.

4.     The rationale for using p65 knock down cells in RNAseq experiments is not clear, as these cells naturally do not respond to TNF stimulation.

5.     The treatment paradigm for experiments in Figure 5B is not clear. In the figure it appears as if cells were pre-treated for 2hrs with CPT, and then stimulated for 0, 1, 8hrs with TNF, but the legend suggests differential treatment of 1 and 8hr timepoints (1h: 2hrs CPT+1hr TNF; 8hrs: 5hrs TNF+2hrs CPT). If this is the case, data cannot be compiled like they are shown.

6.     What is the pharmacodynamics of CPT in cells? Is it still present and active after 8hrs?

7.     What is the impact of CPT on constitutive gene expression?

8.     How does CPT affect gene expression in non-cancer cells? This is important as the authors extrapolate their findings to non-cancerous cells when they speak of relevance of their data for infection risk in cancer patients.

Author Response

Please find enclosed my reply and also figures in the attached rebuttal letter.

Reviewer 3 Report

The authors present an interesting study of the effect of several topoisomerase 1 and 2 inhibitors on TNF and IL1 induced gene expression. The fact that several of these inhibitors are clinically used as anticancer agents makes the study medically relevant. The data clearly point to a major gene-specific role of topoisomerase 1 in the regulation of cytokine induced gene expression. Several additional experiments indicate that topoisomerase 1 inhibition mainly affects early gene induction and does not correlate with inducer strength. Mechanistically, the authors provide evidence for a role of chromatin modification (histone acetylation) and transcription elongation.

Overall this a very well performed study that will be of interest for scientists working in the cancer field but also beyond this.The experiments are well designed and the data support the conclusions made. I only have a minor comment. Since the authors speculate that the reported effects of topoisomerase 1 inhibitors might contribute to increased infection upon cancer treatment and might also be of interest in the context of anti-inflammatory treatment, it may be of interest to analyze if similar effects on TNF/IL1 induced gene expression can be observed on normal cells (= primary immune cells) and not only on cancer cell lines as studied here.

Author Response

(The authors gave the same response as above.)

Round 2

Reviewer 1 Report

Congratulation for the interesting results

Author Response

There was no criticism from this reviewer

Reviewer 2 Report

The authors essentially answered all questions raised by this reviewer.

Residual minor points:

1.     Regarding figure 2F: The authors have now quantified Western Blot data, which confirm that CPT treatment without TNF exposure leads to reduction in IkBa levels and some alteration of p65 phosphorylation. However, this does not appear to be associated with increased nuclear translocation or histone-binding of NF-kB at TNF0. The latter data are only shown in the comments to the reviewer, but I feel that they should be included at least in the supplement, to make the point that CPT alone does not affect p65 NF-kB-dependent transcription, although Western Blot data might suggest this. It would also be helpful to include a discussion about how the authors envision p65 cytosolic retention in absence of IkBa.

Finally, the fractionation experiment also can be used in support of Figure 6 CHIP experiments, as both show less p65 presence on DNA after TNF and CPT, suggesting effects beyond IL8 promoter only.   

2.     Regarding figure 5A: considering the stated pharmacokinetics of the CPT derivative TPT (3hrs half-life), it is not clear if CPT is still in the system at 4 and 6hrs post washout. Consistent with this, +/- CPT values are not significantly different at these timepoints. Due to this caveat, it is advisable to omit these timepoints from the experiment.

3.     Regarding figure 5B: data need to be presented in a different way, as 1h and 8hr treatment is entirely different and cannot be compared. However, the graph suggests that. Biologically, late NF-kB-dependent genes are driven by secondary inducers, like IFNa or b, which may not be affected by CPT. Please discuss.
